# Relationship between postural and kinesthetic awareness, static balance, and weight-bearing asymmetry in individuals with chronic stroke: A cross-sectional study

Mohammad A. ALMohiza[1], Ravi Shankar Reddy[2*]

1 Department of Health Rehabilitation Sciences, College of Applied Medical Sciences, King Saud University, Riyadh, Saudi Arabia, 2 Program Of Physical Therapy, Department of Medical Rehabilitation Sciences, College of Applied Medical Sciences, King Khalid University, Abha, Saudi Arabia

* rshankar@kku.edu.sa

## Abstract

Postural and kinesthetic awareness are essential sensory-perceptual components contributing to balance control and symmetrical weight distribution. In individuals with chronic stroke, deficits in body awareness can impair postural stability and increase asymmetry, yet their precise relationships with balance parameters remain insufficiently explored. This cross-sectional study investigated the associations between postural and kinesthetic awareness and both static balance performance and weight-bearing asymmetry in individuals with chronic stroke. Forty-eight participants who were at least six months post-stroke were assessed using the Postural Awareness Scale (PAS), and joint position sense error was measured via a digital inclinometer. Static balance parameters, including center of pressure (COP), sway area, path length, and sway velocity, were evaluated using a stabilometric force platform. Weight-bearing asymmetry (WBA) was calculated using the two-scale method. Pearson's correlations and multiple linear regression analyses were conducted. Postural awareness was significantly negatively correlated with sway area ($r = -0.53$, $p = 0.001$) and sway velocity ($r = -0.51$, $p = 0.002$), while joint position sense error was positively correlated with these metrics ($r = 0.47–0.49$, $p < 0.01$). Both awareness measures were significantly associated with WBA (PAS: $r = -0.49$; joint position error: $r = 0.48$, $p < 0.01$). Regression analyses identified PAS as a significant predictor of sway area ($\beta = -0.38$, $p = 0.002$) and WBA ($\beta = -0.36$, $p = 0.003$), while joint position sense error significantly predicted sway velocity ($\beta = 0.41$, $p = 0.001$). These findings highlight the independent associations of postural and kinesthetic awareness with postural control and symmetry in individuals with chronic stroke, suggesting their potential relevance for rehabilitation strategies aimed at improving balance and functional stability.

**Data availability statement:** The datasets generated and analyzed during the current study are publicly available in the Zenodo repository at https://doi.org/10.5281/zenodo.16931745. You can cite all versions of the dataset by using the DOI: 10.5281/zenodo.16931745. Further inquiries can be directed to the corresponding authors.

**Funding:** The authors extend their appreciation to the Deanship of Research and Graduate Studies at King Khalid University, KSA, for funding this work through a large research group under grant number RGP.2/336/46.

**Competing interests:** The authors have declared that no competing interests exist.

## Introduction

Balance control is a fundamental aspect of functional mobility and safety, particularly in individuals recovering from neurological conditions such as stroke [1]. Stroke often results in sensorimotor impairments that compromise postural stability, increase the risk of falls, and reduce independence in daily activities [2]. Among the key components influencing postural control are sensory awareness systems—specifically, postural and kinesthetic awareness—which allow individuals to perceive body position and movement in space [3]. Although these sensory domains are interrelated, they represent distinct constructs. Postural awareness refers to the conscious perception of body alignment and positioning in relation to gravitational and internal cues [4]. In contrast, proprioception encompasses unconscious afferent signals from muscles, joints, and tendons that inform the central nervous system about limb position and movement [5]. Verticality perception, meanwhile, reflects the ability to internally estimate one's orientation with respect to vertical or gravitational reference frames and is often evaluated through visual or graviceptive means [6]. Clear delineation of these terms is essential for accurate interpretation of sensory-perceptual impairments following stroke [6]. Disruption of these systems can impair the ability to make accurate postural adjustments during standing and walking, contributing to instability and asymmetrical weight distribution [7]. Understanding the sensory contributions to balance is therefore essential for optimizing rehabilitation strategies that aim to enhance mobility and reduce the risk of falls in stroke survivors.

Previous research has demonstrated that proprioceptive deficits are common after a stroke and are associated with poor balance and walking difficulties [5]. Impaired joint position sense, particularly at the ankle, has been linked to increased sway and slower postural responses during quiet standing [5]. Studies also suggest that stroke survivors often have an altered perception of verticality and poor body orientation, which relate to worse functional outcomes [6]. Birnbaum et al. [7] reported that deficits in perceiving body position were connected to postural asymmetry, while Kochman et al. [8] found that proprioceptive impairments contributed to increased postural sway. Sarıçan et al. [9] further demonstrated that awareness of body orientation influences how well individuals use strategies to maintain balance. Despite this evidence, most studies have examined these sensory functions separately or mainly focused on proprioception without exploring the broader concept of postural awareness as a distinct sensory-perceptual domain. Additionally, there are still limited quantitative studies linking awareness measures to static balance and weight-bearing asymmetry using clinically available tools. Center of pressure (CoP) parameters, including sway area, path length, and sway velocity, are well-established metrics with documented reliability and validity for assessing postural control in individuals with stroke [8]. These measures have been widely used in clinical and research settings due to their sensitivity to balance impairments and responsiveness to functional changes [9].

Although the relationship between sensorimotor function and balance performance in chronic stroke has been studied [10], few investigations have simultaneously examined the roles of both postural and kinesthetic awareness in relation to static

balance control and interlimb weight distribution [11]. This highlights a significant clinical gap, as both types of awareness likely influence how individuals perceive and adjust their posture in response to sensory input and environmental demands [12]. Furthermore, accessible and standardized methods for measuring these sensory aspects, such as the Postural Awareness Scale (PAS) and joint position sense testing, have not been widely used in assessments of post-stroke balance [13,14]. Clarifying the specific roles of postural and kinesthetic awareness in balance and weight-bearing asymmetry could enhance our understanding of sensory-motor integration deficits after stroke and support the development of targeted interventions to address these issues.

The present study aimed to examine the relationship between postural and kinesthetic awareness and (1) static balance performance, and (2) weight-bearing asymmetry in individuals with chronic stroke. Static balance, assessed during quiet standing, was selected as the primary focus because it forms the basis for more complex postural tasks and is particularly sensitive to deficits in sensory awareness. In chronic stroke populations, quiet-standing assessments offer controlled conditions for isolating the contributions of sensory-perceptual variables to balance performance, minimizing task-related motor or cognitive confounders that may influence dynamic measures.

## Materials and methods

### Study design, ethics, and settings

This cross-sectional study was conducted from 12th May 2024–16th March 2025 at the Neurorehabilitation Clinic, Department of Medical Rehabilitation Sciences, CAMS, King Khalid University, Kingdom of Saudi Arabia. Ethical approval was obtained from the Institutional Review Board of King Khalid University (approval number: [REC# 2024−367]), and all procedures adhered to the ethical standards outlined in the Declaration of Helsinki. Prior to participation, each individual provided written informed consent after receiving a detailed explanation of the study's objectives, procedures, potential risks, and benefits. All assessments and data collection were performed within the clinical setting by trained physical therapists under standardized conditions to ensure methodological consistency and participant safety. To minimize assessment bias, the physical therapists conducting the outcome evaluations were blinded to the participants' awareness scores. Additionally, the assessors responsible for administering the PAS and joint position sense measures were not involved in data analysis.

### Participants

Participants were recruited from the outpatient neurorehabilitation services at the Physical Therapy Clinic using a consecutive sampling method. This included both new referrals and individuals receiving follow-up rehabilitation, as long as they met all inclusion criteria. Eligible individuals were identified through clinician referrals and medical record screening based on a confirmed diagnosis of chronic stroke, defined as a cerebrovascular event occurring at least six months prior to enrollment. Stroke diagnosis was verified through neuroimaging reports and clinical documentation according to the World Health Organization (WHO) stroke diagnostic criteria.

Inclusion criteria required participants to be between 40 and 75 years of age, have sufficient cognitive capacity to follow verbal instructions as evidenced by a Mini-Mental State Examination (MMSE) score ≥24, and the ability to stand unassisted for at least 1 minute [15]. The MMSE is a 30-point screening tool widely used to assess orientation, attention, memory, and language [16]. A score ≥24 was used as the cognitive cutoff for inclusion, indicating preserved cognitive function. Only individuals with unilateral hemiparesis and medically stable conditions were included. Participants were also required to have no changes in their rehabilitation regimen for at least four weeks prior to the study. Exclusion criteria comprised the presence of cerebellar or brainstem strokes, bilateral or recurrent strokes, severe musculoskeletal disorders affecting lower limb function, orthopedic surgery within the past six months, visual or vestibular impairments, or any comorbid neurological condition such as Parkinson's disease or multiple sclerosis. All participants underwent a preliminary screening

session to confirm eligibility, during which demographic and clinical data, including MMSE scores and medical history, were recorded. A total of 62 individuals were screened for eligibility. Of these, 14 were excluded due to not meeting the inclusion criteria (n = 9), declining to participate (n = 3), or incomplete baseline data (n = 2), resulting in a final sample of 48 participants. Those meeting the eligibility criteria proceeded to baseline testing, including assessments of postural and kinesthetic awareness, static balance, and weight-bearing asymmetry.

## Variables

**Postural awareness.** Postural awareness, a primary independent variable in this study, was operationally defined as the conscious perception of body position and alignment in space during static stance. It was assessed using the Postural Awareness Scale (PAS) [17], a validated self-report instrument that quantifies body awareness based on internal postural cues. The PAS consists of multiple items rated on a Likert scale, with higher scores indicating greater postural awareness [17]. Specifically, the PAS contains 12 items scored on a 7-point Likert scale, yielding a total score ranging from 12 to 84, with higher scores indicating greater postural awareness [17]. Participants completed the PAS in a quiet, supervised setting to minimize distractions. This measure has been previously employed in studies examining the sensory-perceptual components of balance control and has demonstrated acceptable psychometric properties in neurologically impaired populations. However, it is important to note that the PAS has not been specifically validated in chronic stroke populations. While it offers insight into internal postural habits and body awareness, further psychometric evaluation in this clinical subgroup is warranted to confirm its construct validity and applicability.

**Kinesthetic awareness.** In this study, kinesthetic awareness was assessed using a Dualer IQ Pro Digital Inclinometer (JTECH Medical, USA), a validated instrument frequently employed in clinical proprioceptive testing for joint position sense (JPS) accuracy [18]. The device offers high-resolution angular measurements and has demonstrated reliability in measuring lower limb proprioception in both healthy and neurologically impaired populations. Participants were seated comfortably in an upright position with the knees flexed at 90° and both feet unsupported. The inclinometer sensors were securely attached to the lateral aspect of the tibia and the lateral aspect of the affected foot using elastic Velcro straps, ensuring consistent placement across trials. To eliminate visual compensation, all participants were blindfolded during testing. Auditory distractions were minimized, and instructions were provided in a standardized format to ensure participant understanding and consistency. During each trial, the examiner passively moved the participant's ankle from a neutral position to a target angle of 15° plantarflexion, held the position for 5 seconds, and then returned the foot to neutral. The angle of 15° plantarflexion was selected based on previous clinical proprioception protocols demonstrating its functional relevance for ankle movement during stance and gait [19]. It also allows standardized, reproducible testing without inducing discomfort or fatigue in post-stroke populations. While multi-angle assessments may offer additional data, the current design prioritized simplicity and feasibility in a clinical setting. Participants were then asked to replicate the target angle without assistance. The absolute difference (in degrees) between the target and the replicated angle was recorded as the joint position sense error. Five trials were conducted, each separated by a 10-second rest interval to minimize fatigue and enhance consistency. The mean angular error across all trials was calculated and used for analysis.

## Static balance performance

Static balance performance was a primary outcome variable and was assessed using a TechnoBody® Iso-Free stabilometric force platform (TechnoBody S.r.l., Italy), a computerized posturography system validated for clinical and research use [20]. This system enables real-time acquisition of COP metrics with high precision, allowing detailed analysis of postural control. The platform was calibrated before each testing session according to the manufacturer's guidelines to ensure measurement reliability and consistency. Following a familiarization trial, participants were instructed to stand barefoot on the platform in a standardized position, with their feet shoulder-width apart and aligned

according to the foot placement markers embedded in the platform. Each participant stood quietly for 30 seconds, with arms relaxed by their sides and eyes open, fixating on a visual target placed at eye level, approximately two meters away, to minimize visual tracking variability. During the test, verbal encouragement or physical assistance was avoided to preserve the integrity of spontaneous postural regulation. Safety measures included the presence of a trained physical therapist standing nearby and the use of a safety harness when deemed necessary to prevent falls without restricting natural movement.

Prior to testing, participants were given a clear, standardized explanation of the procedure to ensure understanding and compliance. A familiarization trial was conducted before the actual recording to minimize any novelty-related variability. Each individual completed three valid trials, with at least one minute of rest between trials to prevent fatigue-related bias. The following COP parameters were automatically extracted using the platform's proprietary software: sway area (cm²), sway path length (cm), sway velocity (cm/s), mediolateral (ML) sway amplitude (cm), and anteroposterior (AP) sway amplitude (cm). These metrics provide quantitative indices of balance control, with greater values reflecting poorer postural stability. The average value of the three trials was used for final analysis. These procedures are consistent with established protocols in post-stroke balance assessment literature and ensure the reproducibility and clinical relevance of the findings.

### Weight-bearing asymmetry

Weight-bearing asymmetry (WBA) was defined as the percentage difference in vertical ground reaction forces between the affected and unaffected limbs during static standing [14]. WBA was measured using the two-scale method, where participants stood with one foot on each of two identical calibrated digital scales. The absolute difference in load distribution between limbs was calculated with the formula: WBA (%) = |(Load_Affected – Load_Unaffected)/ Total Body Weight| × 100. Higher WBA values indicated greater asymmetry. This method has been validated as a reliable and clinically feasible way to assess limb loading patterns in individuals after a stroke. The two-scale method was selected due to its simplicity, low cost, and suitability for use in clinical environments lacking access to force platforms. Although it provides lower spatial and temporal resolution than stabilometric systems, it yields reliable estimates of interlimb weight distribution and has demonstrated acceptable validity in prior stroke rehabilitation studies. Its practicality enables broader implementation and supports translation of findings into real-world settings.

### Covariates and descriptive measures

Multiple demographic and clinical variables were collected to characterize the sample and control for potential confounding factors. These included age, sex, time since stroke onset, stroke type (ischemic or hemorrhagic), side of hemiparesis, and MMSE scores for assessing cognitive status. Functional status was measured with the Berg Balance Scale (BBS) and the Fugl-Meyer Assessment for the Lower Extremity (FMA-LE), both standardized, performance-based tools often used in stroke rehabilitation to evaluate balance and motor recovery. These covariates were considered when analyzing primary outcomes and included in descriptive statistical analyses.

### Sample size calculation

The sample size for this study was determined beforehand using G*Power software version 3.1.9.7 (Heinrich Heine University, Düsseldorf, Germany). We used the 'Correlation: Bivariate normal model' and 'Linear multiple regression: Fixed model, R² deviation from zero' analysis types to ensure sufficient statistical power for detecting medium-sized associations between postural and kinesthetic awareness and static balance parameters in individuals with chronic stroke. Based on a Pearson correlation analysis with a two-tailed alpha level set at 0.05, a medium effect size of r = 0.40, and a desired statistical power of 0.80, the minimum required sample size was calculated to be 46 participants. To account for possible attrition or incomplete data, we aimed for a slightly larger sample. For multiple linear regression analyses involving two

predictors (postural and kinesthetic awareness) and one continuous outcome (sway velocity), we assumed a medium effect size of $f^2 = 0.15$, an alpha level of 0.05, and a power of 0.80, which resulted in a required sample size of 43.

## Data analysis

Data were analyzed using IBM SPSS Statistics version 24.0 (IBM Corp., Armonk, NY, USA). Prior to inferential testing, all continuous variables were assessed for normality using the Shapiro-Wilk test and visual inspection of histograms and Q-Q plots, confirming that the data were approximately normally distributed; thus, parametric statistical tests were applied. Representative histograms and Q-Q plots for key outcome variables (sway area, sway velocity, and WBA) are provided in Supplementary File 1 to support the normality assumption. Visual inspection confirmed no substantial skew or kurtosis that would violate parametric assumptions. Descriptive statistics were computed for all demographic, clinical, and outcome variables, and are presented as means and standard deviations for continuous data and as frequencies and percentages for categorical variables. Independent samples *t*-tests were used to compare continuous variables between groups, while chi-square tests were applied for categorical data. To examine associations between postural awareness scores, kinesthetic awareness (measured via joint position sense error), static balance parameters (e.g., sway area, sway velocity), and weight-bearing asymmetry, Pearson's correlation coefficients were calculated. Furthermore, multiple linear regression analyses were conducted to determine the extent to which postural and kinesthetic awareness predicted balance performance and weight-bearing asymmetry, with the predictors entered simultaneously into the model. All statistical tests were two-tailed, and a *p*-value of less than 0.05 was considered indicative of statistical significance. Multicollinearity among predictor variables was assessed using the variance inflation factor (VIF) analysis. VIF values for PAS and joint position sense error were 1.28 and 1.22, respectively, indicating low multicollinearity and acceptable model stability.

## Results

The demographic and clinical characteristics of the participants are shown in Table 1. Descriptive statistics for the sample's demographic and clinical characteristics are presented in Table 1, including age, sex, time since stroke, stroke type, side of hemiparesis, and MMSE scores, along with functional and awareness measures. However, significant differences appeared in functional and sensorimotor measures, with lower Berg Balance Scale scores (p = 0.031) and Fugl-Meyer lower extremity scores (p = 0.045), along with reduced postural awareness as shown by lower PAS values (p = 0.022). Additionally, participants had greater joint position sense error (p = 0.004) and increased weight-bearing asymmetry (p = 0.019), indicating meaningful impairments in balance, proprioception, and postural control.

**Table 1. Demographic and clinical characteristics of participants.**

| Variable | Mean ± SD or n (%) | p-value |
|---|---|---|
| Age (years) | 62.35 ± 9.42 | 0.123 |
| Sex (Male/Female) | 29 (60.42%)/ 19 (39.58%) | 0.541 |
| Time Since Stroke (months) | 18.67 ± 7.25 | 0.089 |
| Stroke Type (Ischemic/Hemorrhagic) | 37 (77.08%)/ 11 (22.92%) | 0.334 |
| Affected Side (Left/Right) | 25 (52.08%)/ 23 (47.92%) | 0.477 |
| MMSE Score | 26.85 ± 1.72 | 0.216 |
| Berg Balance Scale | 42.31 ± 6.98 | 0.031 |
| Fugl-Meyer Lower Extremity Score | 26.43 ± 4.12 | 0.045 |
| Postural Awareness Scale (PAS) | 45.26 ± 8.73 | 0.022 |
| Joint Position Sense Error (°) | 3.92 ± 1.11 | 0.004 |
| Weight-Bearing Asymmetry (%) | 14.65 ± 6.23 | 0.019 |

SD; Standard Deviation; MMSE; Mini-Mental State Examination; PAS; Postural Awareness Scale.

As shown in Table 2 and illustrated in **Fig 1**, postural awareness demonstrated significant negative correlations with all static balance parameters, indicating that higher awareness scores were associated with reduced sway area, path length, velocity, and both mediolateral and anteroposterior sway amplitudes (r ranging from −0.44 to −0.53, p < 0.01). In contrast, greater joint position sense error was positively correlated with these same balance parameters (r ranging from 0.40 to 0.49, p < 0.01), suggesting that impaired proprioceptive accuracy was associated with increased postural instability.

As presented in Table 3 and illustrated in Fig 2, weight-bearing asymmetry showed a significant negative association with postural awareness, with higher PAS scores linked to reduced asymmetry (r = −0.49, β = −0.36, 95% CI: −0.58 to −0.14, p < 0.01). In contrast, greater joint position sense error was positively associated with asymmetry, indicating that impaired proprioceptive accuracy contributed to increased imbalance (r = 0.48, β = 0.34, 95% CI: 0.12 to 0.56, p < 0.01).

As detailed in Table 4 and depicted in Fig 3, multiple linear regression analysis revealed that higher postural awareness was a significant predictor of better balance outcomes, with greater PAS scores associated with reduced COP sway area (β = −0.38, 95% CI: −0.60 to −0.16, p = 0.002) and shorter sway path length (β = −0.36, 95% CI: −0.60 to −0.12, p = 0.004). In contrast, greater joint position sense error independently predicted increased COP sway velocity

**Table 2. Correlation between postural and kinesthetic awareness and static balance parameters.**

| Variable | Mean ± SD | r (PAS) | p-value (PAS) | r (Joint Position Error) | p-value (Joint Position Error) |
|---|---|---|---|---|---|
| COP Sway Area (cm²) | 4.85 ± 1.62 | −0.53 | 0.001 | 0.47 | 0.002 |
| COP Sway Path Length (cm) | 89.42 ± 12.37 | −0.48 | 0.003 | 0.45 | 0.003 |
| COP Sway Velocity (cm/s) | 2.31 ± 0.46 | −0.51 | 0.002 | 0.49 | 0.001 |
| ML Sway Amplitude (cm) | 1.23 ± 0.38 | −0.44 | 0.005 | 0.40 | 0.008 |
| AP Sway Amplitude (cm) | 1.65 ± 0.41 | −0.46 | 0.004 | 0.42 | 0.006 |

COP; Center of Pressure; SD; Standard Deviation; PAS; Postural Awareness Scale; ML; Mediolateral; AP; Anteroposterior.

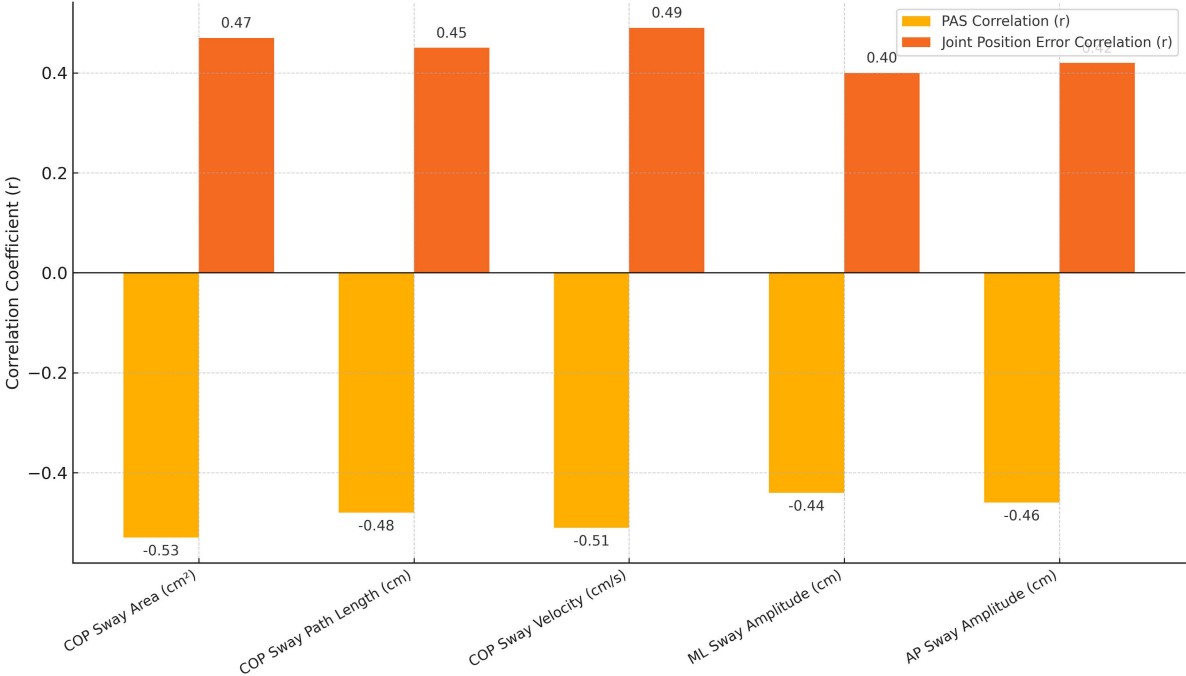

**Fig 1. Correlation of postural and kinesthetic awareness with static balance parameters.**

**Table 3. Association between awareness measures and weight-bearing asymmetry.**

| Predictor Variable | Mean±SD | r (Pearson Correlation with WBA) | p-value | Regression Coefficient (β) | 95% CI for β | p-value (Regression) |
|---|---|---|---|---|---|---|
| Postural Awareness Scale (PAS) | 45.26 ± 8.73 | −0.49 | 0.001 | −0.36 | [-0.58, -0.14] | 0.003 |
| Joint Position Sense Error (°) | 3.92 ± 1.11 | 0.48 | 0.002 | 0.34 | [0.12, 0.56] | 0.004 |

PAS: Postural Awareness Scale; SD: Standard Deviation; WBA: Weight-Bearing Asymmetry; CI: Confidence Interval.

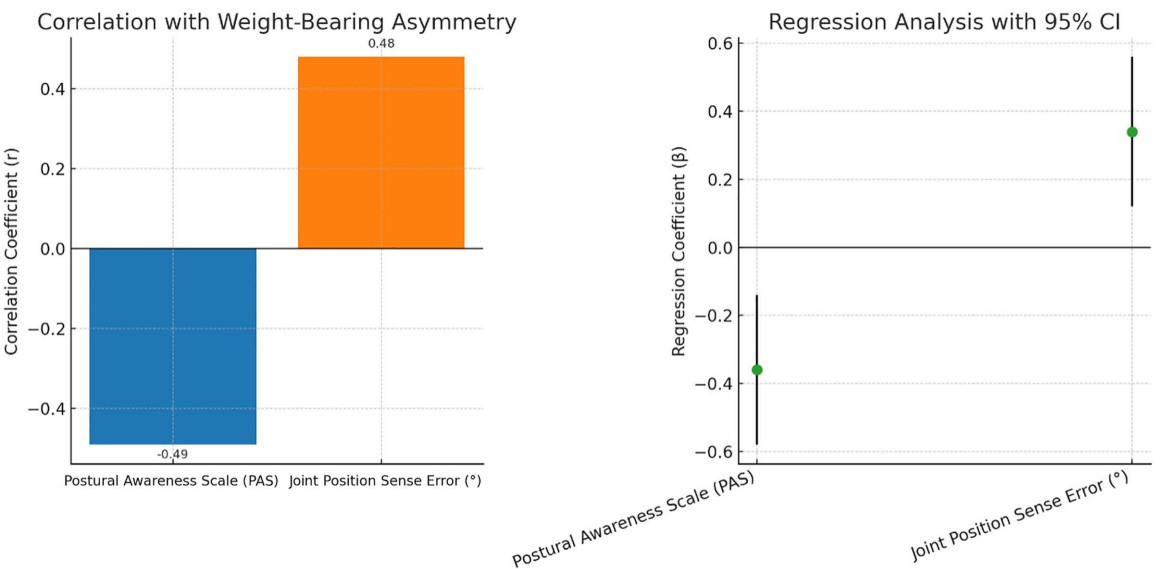

**Fig 2. Association of postural and kinesthetic awareness with weight-bearing asymmetry.**

**Table 4. Multiple linear regression model predicting balance performance from awareness scores.**

| Outcome Variable | Predictor Variable | β Coefficient | Standard Error | 95% CI for β | p-value | R² | Adjusted R² |
|---|---|---|---|---|---|---|---|
| COP Sway Area (cm²) | Postural Awareness Scale (PAS) | −0.38 | 0.11 | [-0.60, -0.16] | 0.002 | 0.28 | 0.26 |
| COP Sway Path Length (cm) | Postural Awareness Scale (PAS) | −0.36 | 0.12 | [-0.60, -0.12] | 0.004 | 0.25 | 0.23 |
| COP Sway Velocity (cm/s) | Joint Position Sense Error (°) | 0.41 | 0.13 | [0.15, 0.67] | 0.001 | 0.31 | 0.29 |

PAS: Postural Awareness Scale; COP: Center of Pressure; CI: Confidence Interval; R²: Coefficient of Determination.

(β = 0.41, 95% CI: 0.15 to 0.67, p = 0.001). The regression models demonstrated acceptable explanatory power (R² = 0.25–0.31), indicating that postural and kinesthetic awareness were statistically associated with variations in static balance performance. To control for potential confounding effects, additional regression models were computed with age, time since stroke, Berg Balance Scale score, and FMA-LE score included as covariates. After adjustment, PAS remained a significant predictor of sway area (β = −0.31, p = 0.007) and path length (β = −0.28, p = 0.011), and joint position sense error continued to significantly predict sway velocity (β = 0.36, p = 0.003), confirming the robustness of the observed associations.

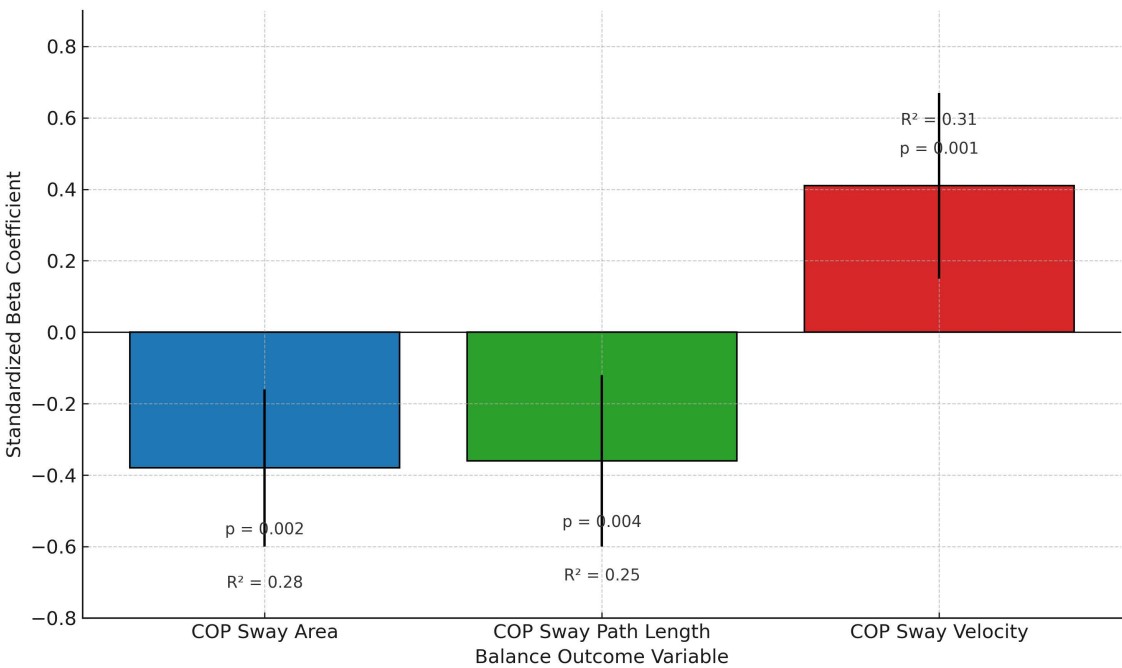

**Fig 3. Regression coefficients illustrating the association between awareness scores and balance performance outcomes.**

## Discussion

The present study aimed to investigate the relationship between postural and kinesthetic awareness with static balance performance and weight-bearing asymmetry in individuals with chronic stroke. The findings demonstrated that lower levels of postural awareness were consistently associated with poorer balance outcomes, while greater deficits in kinesthetic awareness were linked to increased postural instability. Furthermore, both awareness measures were significantly associated with asymmetrical weight distribution, suggesting their potential complementary relevance to stance control.

The observed associations between awareness measures and static balance parameters highlight the crucial role of sensory integration in postural control following stroke [21]. Reduced postural awareness likely impairs the individual's ability to accurately perceive and adjust body alignment, leading to greater displacement of the center of pressure and increased sway in multiple directions [22]. Similarly, diminished kinesthetic accuracy, reflected by higher joint position sense errors, compromises proprioceptive feedback from the lower limbs, thereby limiting the capacity to make timely corrective adjustments during quiet standing [23]. These combined deficits in body awareness and proprioceptive processing contribute to instability in stance, as evidenced by larger sway amplitudes, longer path lengths, and higher sway velocities, underscoring the interdependence of sensory awareness and balance regulation in chronic stroke [23,24]. These findings are consistent with previous research emphasizing the contribution of proprioception and postural awareness to balance performance [25,26]. Xu et al. [25] reported that stroke survivors with impaired proprioception exhibited significantly greater postural sway, supporting the observed link between joint position sense errors and instability. Sarıçan et al. [26] also highlighted that deficits in body orientation awareness negatively influenced balance recovery and stability limits. Similarly, Liu et al. [27] demonstrated that impaired perception of body position was strongly associated with postural asymmetry and instability in individuals with stroke. Collectively, these studies reinforce the current results, confirming that both postural and kinesthetic awareness are integral determinants of balance performance and should be considered key components in post-stroke rehabilitation strategies.

The associations between awareness measures and weight-bearing asymmetry, as well as balance performance, reflect the essential role of sensory feedback in maintaining symmetrical stance and stability after stroke [28,29]. Reduced postural awareness limits an individual's ability to consciously align the body, thereby increasing asymmetrical load distribution between the limbs [29,30]. Conversely, impaired kinesthetic processing, demonstrated by greater joint position sense error, disrupts proprioceptive feedback required for accurate weight transfer, leading to increased asymmetry and instability [23,24]. The regression findings further reinforce these mechanisms by showing that postural awareness strongly predicts reduced sway area and path length, while proprioceptive deficits independently contribute to higher sway velocity, suggesting distinct but complementary influences of body awareness and kinesthetic accuracy on static balance control [23,24]. These results are supported by earlier investigations highlighting the interplay between proprioception, postural perception, and balance regulation in stroke populations. Birnbaum et al. [21] demonstrated that impaired perception of body orientation was associated with greater postural asymmetry, while Cho et al. [10] reported that deficits in proprioceptive function led to poorer weight distribution and balance outcomes. Pinho et al. [22] further showed that altered awareness of body position was linked to reduced stability and impaired compensatory strategies during stance. Collectively, these findings align with the present results, substantiating that both postural and kinesthetic awareness are critical determinants of weight-bearing symmetry and overall postural control in individuals with chronic stroke.

## Clinical significance

The clinical significance of this study lies in demonstrating that both postural and kinesthetic awareness are critical determinants of balance performance and weight-bearing symmetry in individuals with chronic stroke. The findings clearly show that reduced postural awareness is associated with greater sway and asymmetry, while impaired kinesthetic accuracy contributes to increased postural instability and higher sway velocity. These associations highlight the importance of systematically assessing body awareness and proprioceptive function in stroke rehabilitation, as they provide meaningful predictors of postural control deficits. These findings suggest that interventions targeting postural and kinesthetic awareness could be explored in future studies as potential strategies to promote more symmetrical weight distribution and enhance postural stability, though causal inferences require longitudinal or interventional evidence.

## Limitations and area of future research

This study has certain limitations that should be acknowledged. First, although the PAS was used to assess postural awareness, its psychometric validation in chronic stroke populations remains limited. Future research should establish the scale's construct validity and sensitivity within this specific clinical context. The cross-sectional design restricts causal inference, as the observed associations between awareness measures, balance performance, and weight-bearing asymmetry cannot establish directional relationships. The sample size, though adequately powered for the planned analyses, was limited to a single-center cohort, which may affect the generalizability of the findings to broader stroke populations with varying severities and rehabilitation exposures. Additionally, the study focused only on static balance during quiet standing, without accounting for dynamic tasks such as gait or functional mobility, which may provide further insights into real-world postural control. Future research should employ longitudinal or interventional designs to determine whether targeted training of postural and kinesthetic awareness can directly improve balance and symmetry outcomes. Expanding the investigation to include larger, more diverse cohorts, dynamic balance assessments, and neurophysiological correlates will further clarify the mechanistic role of body awareness in post-stroke rehabilitation and inform the development of comprehensive therapeutic strategies.

## Conclusion

This study demonstrated that both postural and kinesthetic awareness are significantly associated with static balance performance and weight-bearing asymmetry in individuals with chronic stroke. Lower postural awareness was linked to

greater sway and asymmetry, while impaired kinesthetic accuracy was associated with increased postural instability, as reflected by higher sway velocity. Regression analyses further confirmed that postural awareness predicted reductions in sway area and path length, whereas joint position sense error independently predicted increased sway velocity. These findings establish postural and kinesthetic awareness as measurable factors contributing to postural control deficits after stroke, underscoring their relevance for comprehensive clinical assessment in this population.

## Author contributions

**Conceptualization:** Mohammad A. ALMohiza, Ravi Shankar Reddy.

**Data curation:** Mohammad A. ALMohiza, Ravi Shankar Reddy.

**Formal analysis:** Mohammad A. ALMohiza, Ravi Shankar Reddy.

**Funding acquisition:** Ravi Shankar Reddy.

**Investigation:** Mohammad A. ALMohiza.

**Methodology:** Mohammad A. ALMohiza, Ravi Shankar Reddy.

**Project administration:** Mohammad A. ALMohiza.

**Resources:** Mohammad A. ALMohiza.

**Software:** Mohammad A. ALMohiza.

**Supervision:** Mohammad A. ALMohiza.

**Visualization:** Mohammad A. ALMohiza.

**Writing – original draft:** Mohammad A. ALMohiza, Ravi Shankar Reddy.

**Writing – review & editing:** Mohammad A. ALMohiza, Ravi Shankar Reddy.

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
