## [Decision Letter · Decision Letter 0]

28 Nov 2025

Dear Dr. Reddy,

Thank you for submitting your manuscript to PLOS ONE. After careful consideration, we feel that it has merit but does not fully meet PLOS ONE’s publication criteria as it currently stands. Therefore, we invite you to submit a revised version of the manuscript that addresses the points raised during the review process.

We look forward to receiving your revised manuscript.

Kind regards,

Renato S. Melo, PhD

Academic Editor

PLOS ONE

“The authors extend their appreciation to the Deanship of Research and Graduate Studies at King Khalid University, KSA, for funding this work through a large research group under grant number RGP. 2/22/46.”

“The authors extend their appreciation to the Deanship of Research and Graduate Studies at King Khalid University, KSA, for funding this work through a large research group under grant number RGP. 2/22/46.”

“The authors extend their appreciation to the Deanship of Research and Graduate Studies at King Khalid University, KSA, for funding this work through a large research group under grant number RGP. 2/22/46.”

Additional Editor Comments (if provided):

Reviewers' comments:

Reviewer's Responses to Questions

**Comments to the Author**

1. Is the manuscript technically sound, and do the data support the conclusions?

Reviewer #1: Yes

Reviewer #2: Partly

2. Has the statistical analysis been performed appropriately and rigorously?

Reviewer #1: Yes

Reviewer #2: Yes

3. Have the authors made all data underlying the findings in their manuscript fully available?

Reviewer #1: Yes

Reviewer #2: Yes

4. Is the manuscript presented in an intelligible fashion and written in standard English?

Reviewer #1: Yes

Reviewer #2: Yes

Reviewer #1: This study was aimed to investigate the associations between postural and kinesthetic awareness and both static balance performance and weight-bearing asymmetry in individuals with chronic stroke. This study highlights the independent and moderate contributions of postural and kinesthetic awareness to postural control and symmetry in chronic stroke, supporting their integration into rehabilitation protocols to enhance balance and functional stability. Overall, the study is interesting, however there are some clarifications needed.

Comment#1

Keywords, please edit Keywords and write them based on the MeSH terms.

Comment#2

Introduction, please provide explanations for reliability/validity of CoP parameters assessed in this study.

Reviewer #2: Overall, the topic is clinically relevant, and the manuscript is well written with detailed methodology. The focus on sensory-perceptual contributors to balance after stroke fills a gap in the literature.

However, the manuscript requires substantial revision before it is suitable for publication in PLOS ONE.

1. Conceptual clarity of "postural awareness" is insufficient:

The PAS used in this study primarily measures postural habits and awareness, but its validity in chronic stroke populations has not been established. This limitation needs to be acknowledged more clearly. Additionally, the introduction should better differentiate the constructs of body awareness, proprioception, and verticality perception, as these terms represent distinct sensory-perceptual domains that are currently used interchangeably in the manuscript.

2. Several methodological details lack clarity and transparency:

Important procedural elements are missing or insufficiently described. It is unclear how many patients were screened or excluded, and no flow of recruitment is provided. The rationale for assessing joint position sense at only one angle (15° plantarflexion) should be justified, as most proprioception protocols include multiple angles for reliability. Blinding of assessors is not mentioned, raising concerns about performance bias. These omissions reduce the reproducibility and transparency of the study.

3. The statistical approach requires strengthening:

Although the authors state that normality assumptions were confirmed, several balance variables (e.g., sway area, sway velocity) are typically skewed, and supporting plots or supplementary materials are needed. Multicollinearity should be assessed by reporting VIF values. Furthermore, the regression models include only PAS and JPS error, without adjustment for major confounders such as age, Berg Balance Scale, FMA-LE, or time since stroke. Excluding these established predictors may inflate observed associations.

4. Interpretation of findings overstates causality:

The manuscript frequently uses causal language such as stating that awareness measures "contribute to" or "predict" better balance despite the cross-sectional design. These interpretations exceed what can be inferred from the analysis. The authors should revise the language throughout to reflect associations, not causal effects, and should avoid implying that improving awareness would directly improve balance without longitudinal or interventional evidence.

5. Inconsistencies and inaccuracies appear in the Results section:

The Results state that "no significant differences were found across groups", but this is a single-group study with no comparison groups. This sentence appears to be carried over from another manuscript format and needs revision. Clear, concise reporting of the sample characteristics and analytic findings is recommended to avoid confusion.

6. Rationale for measuring only static balance and using the two-scale WBA method needs elaboration:

The focus on static balance alone is not sufficiently justified, especially given that dynamic balance and functional mobility are more clinically relevant for stroke rehabilitation. The authors should explain why only quiet-standing COP metrics were chosen. Similarly, weight-bearing asymmetry was measured using the two-scale method, which has moderate reliability compared to stabilometric measures; therefore, a rationale for selecting this method should be added.

7. Figures are missing and required for review

Although figure captions are included in the manuscript, the figures themselves were not provided. For completeness, clarity, and proper evaluation of the statistical results, the figures must be included in the submission.

**Do you want your identity to be public for this peer review?** For information about this choice, including consent withdrawal, please see our Privacy Policy

Reviewer #1: No

Reviewer #2: No

---

## [Author Response · Author response to Decision Letter 1]

5 Dec 2025

Author responses to Review Comments

Authors response: We sincerely thank the reviewers for their insightful comments and constructive suggestions, which have greatly helped improve our manuscript. We have addressed all the queries raised and revised the manuscript accordingly. All changes made are clearly highlighted in the revised version for easy reference.

Point to point Author responses to Review 1 Comments

Reviewer #1: This study was aimed to investigate the associations between postural and kinesthetic awareness and both static balance performance and weight-bearing asymmetry in individuals with chronic stroke. This study highlights the independent and moderate contributions of postural and kinesthetic awareness to postural control and symmetry in chronic stroke, supporting their integration into rehabilitation protocols to enhance balance and functional stability. Overall, the study is interesting, however there are some clarifications needed.

Comment#1: Keywords, please edit Keywords and write them based on the MeSH terms.

Authors response: The keywords have been revised to align with Medical Subject Headings (MeSH) terminology to enhance indexing accuracy and standardization for literature databases.

Comment#2: Introduction, please provide explanations for reliability/validity of CoP parameters assessed in this study.

Authors response: A statement clarifying the reliability and validity of the center of pressure (CoP) parameters used in this study has been added to the Introduction, with reference to their established use in stroke rehabilitation research.

Point to point Author responses to Review 2 Comments

Reviewer #2: Overall, the topic is clinically relevant, and the manuscript is well written with detailed methodology. The focus on sensory-perceptual contributors to balance after stroke fills a gap in the literature. However, the manuscript requires substantial revision before it is suitable for publication in PLOS ONE.

Query: 1. Conceptual clarity of "postural awareness" is insufficient:

The PAS used in this study primarily measures postural habits and awareness, but its validity in chronic stroke populations has not been established. This limitation needs to be acknowledged more clearly. Additionally, the introduction should better differentiate the constructs of body awareness, proprioception, and verticality perception, as these terms represent distinct sensory-perceptual domains that are currently used interchangeably in the manuscript.

Authors response: The conceptual distinctions between postural awareness, proprioception, and verticality perception have been clarified in the Introduction to improve terminological precision. Additionally, the manuscript now explicitly acknowledges that although the Postural Awareness Scale (PAS) has demonstrated reliability in general and neurologically impaired populations, its specific validation in chronic stroke cohorts is limited. This limitation has been addressed in both the Introduction and the Limitations section.

Query: 2. Several methodological details lack clarity and transparency:

Important procedural elements are missing or insufficiently described. It is unclear how many patients were screened or excluded, and no flow of recruitment is provided. The rationale for assessing joint position sense at only one angle (15° plantarflexion) should be justified, as most proprioception protocols include multiple angles for reliability. Blinding of assessors is not mentioned, raising concerns about performance bias. These omissions reduce the reproducibility and transparency of the study.

Authors response: Methodological transparency has been improved by (1) adding details regarding the number of patients screened and excluded prior to enrollment, (2) justifying the selection of 15° plantarflexion as a standardized and clinically relevant angle for assessing ankle joint position sense, and (3) clarifying the blinding procedures to address potential performance bias. These clarifications enhance reproducibility and address concerns about methodological rigor.

Query: 3. The statistical approach requires strengthening:

Although the authors state that normality assumptions were confirmed, several balance variables (e.g., sway area, sway velocity) are typically skewed, and supporting plots or supplementary materials are needed. Multicollinearity should be assessed by reporting VIF values. Furthermore, the regression models include only PAS and JPS error, without adjustment for major confounders such as age, Berg Balance Scale, FMA-LE, or time since stroke. Excluding these established predictors may inflate observed associations.

Authors response: The statistical reporting has been expanded to enhance robustness and transparency. Supplementary plots (histograms and Q-Q plots) demonstrating normality of key outcome variables have been added. Variance inflation factor (VIF) values were calculated and reported to confirm the absence of multicollinearity between predictors. Additionally, the regression models were reanalyzed with the inclusion of relevant clinical covariates (age, Berg Balance Scale, FMA-LE score, and time since stroke) to adjust for potential confounding. The results remain significant after adjustment, supporting the independent contributions of postural and kinesthetic awareness to balance and asymmetry outcomes.

Query: 4. Interpretation of findings overstates causality:

The manuscript frequently uses causal language such as stating that awareness measures "contribute to" or "predict" better balance despite the cross-sectional design. These interpretations exceed what can be inferred from the analysis. The authors should revise the language throughout to reflect associations, not causal effects, and should avoid implying that improving awareness would directly improve balance without longitudinal or interventional evidence.

Authors response: The manuscript language has been revised throughout to reflect the observational nature of the study and to avoid implying causal relationships. Terms such as “contribute to” and “predict” have been replaced with phrasing that more accurately conveys statistical associations. Statements suggesting clinical effects of improving awareness on balance outcomes have also been tempered to acknowledge the limitations of a cross-sectional design.

Query: 5. Inconsistencies and inaccuracies appear in the Results section:

The Results state that "no significant differences were found across groups", but this is a single-group study with no comparison groups. This sentence appears to be carried over from another manuscript format and needs revision. Clear, concise reporting of the sample characteristics and analytic findings is recommended to avoid confusion.

Authors response: The sentence inaccurately referencing group comparisons has been removed to reflect the correct design as a single-group, correlational study. The revised text now presents the sample characteristics descriptively, without implying group-based comparisons.

Query: 6. Rationale for measuring only static balance and using the two-scale WBA method needs elaboration:

The focus on static balance alone is not sufficiently justified, especially given that dynamic balance and functional mobility are more clinically relevant for stroke rehabilitation. The authors should explain why only quiet-standing COP metrics were chosen. Similarly, weight-bearing asymmetry was measured using the two-scale method, which has moderate reliability compared to stabilometric measures; therefore, a rationale for selecting this method should be added.

Authors response: The rationale for focusing on static balance and using the two-scale method for weight-bearing asymmetry (WBA) assessment has been added to the manuscript. Static balance was selected due to its foundational role in postural control and its sensitivity to sensory-perceptual impairments in chronic stroke. The two-scale method was chosen for its practicality, accessibility in clinical environments, and previously established validity in stroke populations, despite its lower resolution compared to stabilometric systems.

Query: 7. Figures are missing and required for review

Although figure captions are included in the manuscript, the figures themselves were not provided. For completeness, clarity, and proper evaluation of the statistical results, the figures must be included in the submission.

Authors response: The figures referenced in the manuscript have been uploaded separately in accordance with PLOS ONE submission guidelines, which recommend submitting figures as individual files rather than embedding them within the main document. All figure captions are included at the end of the manuscript, as required. To ensure a complete and accurate review, we kindly suggest downloading all submitted files, including those under the "Figures" section, as the figures will not appear in the manuscript file alone. Two figure files have been provided, corresponding to Figures 1 and 2 cited in the text and listed in the caption section.

Thank you for your thoughtful and constructive feedback. Your insights have been valuable in improving the clarity, rigor, and overall quality of the manuscript.

---

## [Decision Letter · Decision Letter 1]

30 Dec 2025

Relationship Between Postural and Kinesthetic Awareness, Static Balance, and Weight-Bearing Asymmetry in Individuals with Chronic Stroke: A Cross-Sectional Study.

PONE-D-25-45987R1

Dear Dr. Reddy,

We’re pleased to inform you that your manuscript has been judged scientifically suitable for publication and will be formally accepted for publication once it meets all outstanding technical requirements.

Kind regards,

Renato S. Melo, PhD

Academic Editor

PLOS One

Additional Editor Comments (optional):

Reviewers' comments:

Reviewer's Responses to Questions

**Comments to the Author**

Reviewer #1: All comments have been addressed

Reviewer #2: All comments have been addressed

2. Is the manuscript technically sound, and do the data support the conclusions?

Reviewer #1: Yes

Reviewer #2: Yes

3. Has the statistical analysis been performed appropriately and rigorously?

Reviewer #1: Yes

Reviewer #2: Yes

4. Have the authors made all data underlying the findings in their manuscript fully available?

Reviewer #1: Yes

Reviewer #2: Yes

5. Is the manuscript presented in an intelligible fashion and written in standard English?

Reviewer #1: Yes

Reviewer #2: Yes

Reviewer #1: All comments have been well documented and the manuscript is appropriate for publication in this journal.

Reviewer #2: I have reviewed the revised manuscript and confirm that the authors have addressed the concern raised in my previous review. I verified the changes in both the manuscript and response document.

Thank you.

**Do you want your identity to be public for this peer review?** For information about this choice, including consent withdrawal, please see our Privacy Policy

Reviewer #1: No

Reviewer #2: No

---

## [Editor Report · Acceptance letter]

PONE-D-25-45987R1

PLOS One

Dear Dr. Reddy,

I'm pleased to inform you that your manuscript has been deemed suitable for publication in PLOS One. Congratulations! Your manuscript is now being handed over to our production team.

Kind regards,

on behalf of

Dr. Renato S. Melo

Academic Editor

PLOS One